# The mRubyFT Protein, Genetically Encoded Blue-to-Red Fluorescent Timer

**DOI:** 10.3390/ijms23063208

**Published:** 2022-03-16

**Authors:** Oksana M. Subach, Aleksandr Tashkeev, Anna V. Vlaskina, Dmitry E. Petrenko, Filipp A. Gaivoronskii, Alena Y. Nikolaeva, Olga I. Ivashkina, Konstantin V. Anokhin, Vladimir O. Popov, Konstantin M. Boyko, Fedor V. Subach

**Affiliations:** 1Complex of NBICS Technologies, National Research Center “Kurchatov Institute”, 123182 Moscow, Russia; subach_om@nrcki.ru (O.M.S.); vlaskina_av@nrcki.ru (A.V.V.); dmitry.e.petrenko@gmail.com (D.E.P.); nikolaeva_ay@nrcki.ru (A.Y.N.); ivashkina_oi@nrcki.ru (O.I.I.); vpopov@fbras.ru (V.O.P.); 2Unit of Animal Genomics, GIGA Research Center, University of Liège, 4000 Liege, Belgium; tashkeev.alex@gmail.com; 3Bach Institute of Biochemistry, Research Centre of Biotechnology of the Russian Academy of Sciences, 119071 Moscow, Russia; gaivphilippp09@gmail.com (F.A.G.); kmb@inbi.ras.ru (K.M.B.); 4Faculty of Biology, M.V. Lomonosov Moscow State University, 119991 Moscow, Russia; 5Laboratory for Neurobiology of Memory, P.K. Anokhin Research Institute of Normal Physiology, 125315 Moscow, Russia; k.anokhin@gmail.com; 6Institute for Advanced Brain Studies, M.V. Lomonosov Moscow State University, 119991 Moscow, Russia; 7Moscow Institute of Physics and Technology, Institutsky Lane 9, Dolgoprudny, 141700 Moscow, Russia

**Keywords:** genetically encoded blue-to-red fluorescent timers, protein engineering, fluorescence imaging, fluorescent protein, mRubyFT, RubyFT#11-9, X-ray structure

## Abstract

Genetically encoded monomeric blue-to-red fluorescent timers (mFTs) change their fluorescent color over time. mCherry-derived mFTs were used for the tracking of the protein age, visualization of the protein trafficking, and labeling of engram cells. However, the brightness of the blue and red forms of mFTs are 2–3- and 5–7-fold dimmer compared to the brightness of the enhanced green fluorescent protein (EGFP). To address this limitation, we developed a blue-to-red fluorescent timer, named mRubyFT, derived from the bright mRuby2 red fluorescent protein. The blue form of mRubyFT reached its maximum at 5.7 h and completely transformed into the red form that had a maturation half-time of 15 h. Blue and red forms of purified mRubyFT were 4.1-fold brighter and 1.3-fold dimmer than the respective forms of the mCherry-derived Fast-FT timer in vitro. When expressed in mammalian cells, both forms of mRubyFT were 1.3-fold brighter than the respective forms of Fast-FT. The violet light-induced blue-to-red photoconversion was 4.2-fold less efficient in the case of mRubyFT timer compared to the same photoconversion of the Fast-FT timer. The timer behavior of mRubyFT was confirmed in mammalian cells. The monomeric properties of mRubyFT allowed the labeling and confocal imaging of cytoskeleton proteins in live mammalian cells. The X-ray structure of the red form of mRubyFT at 1.5 Å resolution was obtained and analyzed. The role of the residues from the chromophore surrounding was studied using site-directed mutagenesis.

## 1. Introduction

Fast-FT, Medium-FT, and Slow-FT fluorescent timers (FTs) were previously developed based on the mCherry red fluorescent protein [1]. We call these FTs “true FTs”, since the intermediate blue fluorescent form in these FTs forms first and completely converts into the red form over time. The complete blue-to-red transition in “true FTs” is an advantage for calculating the exact time of the maturation of proteins after the pulsed start of expression in biological objects. In addition, these FTs are monomeric proteins. This property allows the fluorescence labeling of the individual proteins. Fast-FT, Medium-FT, and Slow-FT fluorescent timers have different maturation times of blue and red forms, which allow their application in cellular processes with different time characteristics. At 37 °C, blue fluorescence maxima for Fast-FT, Medium-FT, and Slow-FT were observed in 0.25, 1, and 10 h, respectively. Half-maxima of red fluorescence for these FTs were reached in 7, 4, and 28 h, respectively. The timer behavior of the timers was similar in both bacterial cells and insect and mammalian cells. The main drawback of the available “true FTs” is their low brightness, especially of the red form, since the blue and red forms of these timers are 2–3- and 5–7-fold less bright than EGFP, respectively.

All other developed FTs are “pseudo-FTs” [2,3]. In “pseudo-FTs”, both fluorescent forms mature independently of each other, and the fluorescence of both forms reaches a plateau and remains at this level over time. The timer properties of “pseudo-FTs” are determined by the different maturation times of these two forms. For biological applications, “true FTs” based on mCherry are more convenient, since in later times, one of the forms completely disappears, unlike pseudo-timers, in which both forms are preserved in later times.

Earlier, the crystal structure of the Fast-FT timer with a resolution of 1.81 Å was determined [4]. Structural data, in combination with the site-directed mutagenesis of Fast-FT, showed that the transition from the blue fluorescent form to red during the maturation of the timer is due to the oxidation of the Tyr67 Cα-Cβ bond, which is a part of the timer chromophore. These data also found the critical role of the amino acid residues in the chromophore’s surrounding in the delayed oxidation of the Tyr67 Cα-Cβ bond compared to the original mCherry protein, and these residues were suggested to be responsible for the timer behavior.

In this article, we developed a blue-to-red “true FT”, called mRubyFT, based on the bright mRuby2 red fluorescent protein (RFP). We characterized the main properties of the purified mRubyFT protein in vitro. We compared the brightness and efficiency of blue-to-red photoconversion for the mRubyFT protein with the respective characteristics for the Fast-FT in mammalian cells. The blue-to-red transition for mRubyFT was also assessed in mammalian cells. We obtained and described the X-ray crystal structure of the mRubyFT protein in its red form. Finally, using site-directed mutagenesis, we characterized the impact of the mRubyFT chromophore’s surrounding on the timer properties of the mRubyFT protein.

## 2. Results and Discussion

### 2.1. Developing Blue-to-Red Fluorescent Timer Based on mRuby2 RFP in E. coli

To create the genetically encoded blue-to-red fluorescent timer, the mRuby2 RFP was first subjected to random mutagenesis followed by screening for the blue and red forms in the bacterial system. The mRuby2 was chosen as a template for the development of the blue-to-red FTs, since it has a 2.7-fold larger brightness compared to the mCherry protein, which was used for the development of the Fast-FT timer [5]. The gene of the bright mRuby2 RFP was cloned into an inducible arabinose system. Then, we subjected the original mRuby2 gene to nine rounds of random mutagenesis followed by screening on Petri dishes in the bacterial system. After each round, we carried out an analysis of the resulting libraries on Petri dishes containing 0.2% arabinose, the addition of which caused the expression of the timers. We marked the bluest/non-red colonies 18 h after plating the bacteria on Petri dishes in the presence of an arabinose inducer, and selected the most red/non-blue colonies 72 h after plating. After each round, 2–3 of the best mutants were subjected to the next round of random mutagenesis. The last two rounds of random mutagenesis did not lead to a noticeable improvement in the brightness of both forms, so we decided that we had reached saturation by these criteria and stopped the optimization of the timer. We called the found best timer RubyFT#11-9. According to the alignment of the amino acid sequences, RubyFT#11-9 had seven mutations compared to the mRuby2 original template (Figure 1 and Appendix A). Only one M167R mutation was internal relative to the β-barrel; thus, we suggested that the M167R mutation was sufficient to stabilize the blue form of the timer. We characterized the spectral properties of both forms for RubyFT#11-9 (Appendix A). However, the examination of the maturation kinetics of the purified RubyFT#11-9 revealed that about 28% of the blue form was preserved even after the fluorescence of the red form reached a plateau (Appendix A). In addition, RubyFT#11-9 was prone to dimerization (Appendix A). These drawbacks of the RubyFT#11-9 variant prompted us to continue the development of the blue-to-red fluorescent timer, which would have complete blue-to-red transition and would be monomeric.

To address the drawbacks of the RubyFT#11-9 timer, we again subjected the original mRuby2 protein to rational mutagenesis followed by screening for the blue-to-red variants on Petri dishes in bacteria, and on the hemi-native polyacrylamide gel electrophoresis (PAGE) on pure proteins to assess their monomeric state. We generated an overlap library of mRuby2 with mutations at positions 65, 148, 165, 167, 220, and 224 (Figure 1 and Appendix A). The positions and residues in these positions for rational mutagenesis were chosen according to the mutations found in the red proteins, in which maturation occurs through an intermediate blue form [6,7,8]. Moreover, these residues and positions for directed mutagenesis were found during the development of earlier blue-to-red fluorescent timers based on mCherry [1]. The availability of X-ray structural data for the mRuby RFP template [9] and the mTagBFP blue fluorescent protein [8], which has a stabilized blue form, also helped to suggest positions in the mRuby2 protein template for rational mutagenesis. During the screening of the rational library under Leica stereomicroscope, we identified mutants with a blue-to-red timer-like phenotype, i.e., in which the blue form turned into red over time and the blue-to-red transition was complete. Screening on Petri dishes was performed as described above. Several of the brightest mutants with a timer-like phenotype were used as templates for one round of the random mutagenesis, followed by screening on Petri dishes as described above. The several variants with the brightest blue and red forms in bacterial streaks were expressed in E. coli bacterial cells, then purified on a Ni-NTA resin and their spectral properties and brightness of the blue and red forms were estimated. The purified mRubyFT variants were also loaded onto PAGE to assess their oligomeric state. The round of random mutagenesis did not lead to a noticeable improvement in the brightness of either form, so we decided that we had reached saturation by these criteria and stopped optimizing the monomeric timer. According to the monomeric state and largest brightness, the brightest monomeric variant RubyFT#14f, hereinafter referred to as mRubyFT, was finally selected (Figure 1). The mRubyFT had six mutations relative to the original mRuby2 template (Figure 1). Mutations N129D and L231R were external to β-can and they were probably important for the mRubyFT protein folding. Mutations M65L, N148S, Q220L, and A224S were inside the β-barrel. We suggest that they were responsible for the timer-like behavior of the mRubyFT timer. The mutation M65L was important because it was a part of the chromophore tripeptide. The amino acid residues at positions 148, 220, and 224 were located close to the chromophore and were most likely needed to optimize the brightness of the blue and red forms and temporal characteristics. Compared to mCherry-based FTs, positions 65 and 220 were found in the mRubyFT blue-to-red fluorescent timer for the first time. Thus, the monomeric blue-to-red mRubyFT fluorescent timer was finally engineered from the mRuby2 protein and the new amino acid positions 65 and 220 were found in mRubyFT that determined its timer properties.

### 2.2. In Vitro Characterization of Purified mRubyFT Timer

First, we characterized the spectral properties and molecular brightness of the developed mRubyFT timer and compared them to the characteristics of the Fast-FT timer (Figure 2 and Table 1). The blue and red forms of the mRubyFT had absorption/excitation/emission maxima at 406/408/457 and 577/582/624 nm, respectively (Figure 2a,b). The emission of the blue form of mRubyFT was 9 nm blue-shifted compared to the blue-emission for Fast-FT (Table 1). The emission of the red form of mRubyFT was 18 nm red-shifted compared to the red form of Fast-FT (Table 1). According to the acid/alkaline denaturation method of extinction coefficient determination, the blue and red forms of the mRubyFT timer were 4.1-fold brighter and 1.3-fold dimmer than the respective forms for the Fast-FT timer, created earlier on the basis of the mCherry RFP and having the brightest red form (Table 1). If the extinction coefficient was determined relative to the absorption at 280 nm, the blue and red forms of the mRubyFT timer were 3.1- and 1.5-fold brighter than the respective forms for the Fast-FT timer (Table 1). Such a discrepancy in the brightness may be attributed to the different maturation efficiency of the Fast-FT and mRubyFT timers in bacterial cells.

**Table 1 ijms-23-03208-t001:** In vitro properties of the purified blue-to-red timer mRubyFT. ^a^—extinction coefficients for red forms were determined by alkaline denaturation method or relative to the absorption at 280 nm (*); extinction coefficients for blue forms were determined by acid denaturation method or relative to the absorption at 280 nm (*). ^b^—quantum yields (QYs) for blue and red forms were determined relative to mTagBFP2 (QY of 0.64) and mCherry (QY of 0.22), respectively. ^c^—brightness was calculated as a product of quantum yield and extinction coefficient relative to the brightness of EGFP protein (QY of 0.6 and extinction coefficient of 56,000 M^−1^ cm^−1^). ^d^—characteristic times for the blue- and red forms of Fast-FT and mRubyFT correspond to the maximum of the blue fluorescence and half of the red fluorescence, respectively, at 37 °C. ^e^—data from references [1,4]. The blue and red font color reflects the fluorescence color of the respective blue and red form.

Timer	Form	Abs, Ex/Em (nm)	ε (mM^−1^ cm^−1^) ^a^	QY ^b^	Brightness vs. EGFP ^c^(%)	Characteristic times, h ^d^	p*K*_a_
Fast-FT ^e^	Blue	403, ND/466	49.7 (18.4) *	0.3	44 (16) *	0.25	2.8
	Red	583, ND/606	75.0 (19.1) *	0.09	20 (5.1) *	7.1	4.1
mRubyFT	Blue	406, 408/457	96.8 (26.0) *	0.63	181 (49) *	5.7	3.9 ± 0.5
	Red	577, 582/624	60.0 (29.6) *	0.086	15 (7.6) *	15	4.5 ± 0.1

To characterize the timer characteristics of the mRubyFT timer, the time of maximum fluorescence for the blue form and the half-time of maturation for the red form of the purified timer were determined (Figure 2c). For this purpose, an overnight culture was induced for 1 h with a lack of oxygen. Then, the bacteria were lysed and the protein was purified on Ni-NTA agarose for 20–30 min. Next, the kinetics of maturation for the blue and red forms were recorded on a SOLAR spectrofluorometer at 37 °C. Figure 2c illustrates a time dependence of the blue and red fluorescence of mRubyFT timer and shows the characteristic times for the blue and red forms. At 37 °C, the maximal fluorescence of the blue form of the mRubyFT timer was found at 5.7 h (Figure 2c and Table 1). At 37 °C, the maturation half-time of the red form of the mRubyFT timer was observed at 15 h (Figure 2c and Table 1). At 37 °C, the blue form of mRubyFT completely transformed into the red form after about 45 h. Thus, the characteristic times of the blue and red forms of the mRubyFT timer were most consistent with those for Slow-FT (9.8 and 28 h, respectively) [1]. We also calculated the red-to-blue ratio according to the red and blue fluorescence time dependences normalized to 100 (Figure 2c). During 48 h maturation of the mRubyFT protein at 37 °C, the red-to-blue fluorescence ratio changed from 0 to the value of 39. Storage of the purified mRubyFT protein at 4 °C for one week resulted in a decrease in the blue form absorption by 22% and a 2.4-fold increase in the red form absorption (Appendix A). Hence, blue-to-red transition occurs even at 4 °C storage, which hinders the in vivo labeling of two neuronal populations activated in episode A and B [10].

We then studied the pH stability of both forms of the mRubyFT timer (Figure 2d and Table 1). Both the blue and red forms of mRubyFT were resistant to pH changes with pKa values of 3.9 and 4.5, respectively. mRubyFT was less pH stable compared to Fast-FT (Table 1). The high pH stability of mRubyFT fluorescence makes it a good candidate for studies in acidic cellular compartments such as lysosomes.

To characterize the oligomeric properties of the mRubyFT timer in vitro on purified protein, we loaded the mRubyFT protein onto a semi-native polyacrylamide gel electrophoresis (Appendix A) and fast protein liquid chromatography (FPLC) (Figure 2e and Appendix A). The mRubyFT timer was a monomer both on PAGE and FPLC. The mRubyFT timer also crystallized as a monomer (see Section 2.5). Indeed, according to the alignment of the amino acid sequences (Figure 1), the mRubyFT timer does not contain external mutations in AB and CD interfaces leading to protein oligomerization [8]. According to the X-ray structure of the mRubyFT, these external mutations should stabilize the monomeric state and perturb potential AC and AD inter-subunit interfaces (see Section 2.5). Hence, mRubyFT is monomeric and can be applied for the individual protein labeling.

The blue-to-red mCherry-derived FTs were prone to photoconversion with high-intensity violet light, which caused light-induced transition from the blue form to the fluorescent red form [1]. To test the blue-to-red photoconversion in the case of mRubyFT, it was illuminated with 405 nm LED array and the absorption and fluorescence spectra were recorded. The illumination of the purified mRubyFT timer with 405 nm light resulted in a decrease in the absorption peak at 406 nm, and the appearance of the new absorption peak at 624 nm corresponded to the far-red form (Figure 2f). The 624 nm absorbing far-red form was not fluorescent. Hence, unlike the mCherry-derived FTs, the violet light photoconverted the blue form of the mRubyFT timer not to the fluorescent red form, but to the non-fluorescent far-red form; these data suggest that the photobleaching of the blue form of the mRubyFT timer should not result in the effective appearance of the undesired red fluorescent form during imaging of the mRubyFT in mammalian cells. The light-induced formation of the far-red chromophore with excitation/emission maxima peaked at 636/662 nm, respectively, was earlier observed for a PSmOrange fluorescent protein that was photo-switched from the orange form to the far-red form using blue light [11].

### 2.3. Behavior of the mRubyFT Timer in Cultured Mammalian Cells

To evaluate the folding efficiency and brightness of the mRubyFT timer in mammalian cells in comparison with the control Fast-FT, the selected timers were cloned into a mammalian vector, pAAV-CAG-FT-P2A-EGFP, transiently expressed in HeLa cells and imaged 24 and 72 h after transfection using a confocal microscope. The pAAV-CAG-FT-P2A-EGFP vector provides equimolar expression of timers and EGFP after cleavage of the fusion protein FT-P2A-EGFP at the self-cleavable peptide P2A. At 24 and 72 h after the transfection of the HeLa cells, the brightness of the blue and red forms of the timers were evaluated, respectively, according to the analysis of the confocal images of the cells (a typical image is shown in Figure 3a). The blue and red forms of the mRubyFT timer were 1.27- (*p* < 0.001) and 1.28-fold (*p* = 0.0011) brighter, respectively, than the corresponding forms of the Fast-FT timer (Figure 3b and Table 2). Hence, both forms of the mRubyFT timer had 1.3-fold higher brightness in HeLa cells compared to the brightness of the respective forms for the control Fast-FT timer.

For the mCherry-derived FTs, blue-to-red photoconversion using violet light was noted before [1]. Hence, we compared the efficiency of the blue-to-red photoconversion with violet light for mRubyFT and Fast-FT timers expressing in live HeLa cells as described above. The irradiation with high-power 395/25 nm light (metal halide lamp, 0.9 mW/cm^2^ power measured before 60x oil objective lens) for 1 min of the HeLa cells expressing Fast-FT timer resulted in photobleaching of the blue form and appearance of the red form, with mean ΔF/F values of −1.9 and 4.0, respectively (Figure 3c–e). The illumination of the mRubyFT expressing in HeLa cells using the same conditions photobleached the blue form and increased the fluorescence of the red form with mean ΔF/F values of −2.0 and 0.99, respectively (Figure 3c–e). To calculate the efficiency of the blue-to-red photoconversion, we normalized the ΔF/F values for the red form to the ΔF/F values for the blue form (Figure 3d). The efficiency of the blue-to-red photo-transformation for the mRubyFT timer (mean value of 2.1) was 4.2-fold less than the respective efficiency (mean value of 0.50) for the control Fast-FT timer (Figure 3d). The inefficient blue-to-red photoconversion of the mRubyFT is probably related to the formation of the 624 nm absorbing non-fluorescent far-red species (Figure 2f). In contrast to the notable blue-to-red photoconversion of mRubyFT in mammalian cells, we did not see the formation of the red species during the irradiation of the purified mRubyFT timer in vitro (Figure 2f); this difference is probably related to the different composition of the medium during photoconversion in vitro and in the cytosol of the cells where different electron acceptors are present [12]. Hence, mRubyFT in the cytosol of mammalian cells is photoconverted with violet light from the blue fluorescent color to the red one; however, the efficiency of the blue-to-red photoconversion is 4.2-fold less compared to the efficiency of the same photoconversion of the mCherry-derived Fast-FT timer.

To assess the timer behavior of the mRubyFT protein in mammalian cells, we transiently expressed mRubyFT timer in HEK293T cells under the control of the Tet-Off system at 37 °C and registered changes of their blue and red fluorescence over time [1]. At 16 h after transfection, the expression of the mRubyFT timer was stopped by the addition of doxycycline, and the blue and red fluorescence was registered using confocal imaging (Figure 4). The blue fluorescence of mRubyFT reached its maximum at 8 h and gradually decreased afterwards (Figure 4). The red fluorescence of mRubyFT reached its maximum at 73 h (Figure 4). The maximum of the red-to-blue ratio was observed at 48 h (Figure 4). Hence, the mRubyFT protein preserved its blue-to-red fluorescence timer behavior, when expressed in mammalian cells.

### 2.4. Behavior of mRubyFT in Fusions with Cytoskeleton Proteins in Mammalian Cells

To assess the applicability of the mRubyFT timer for protein labeling and the absence of the tendency to form aggregates, mRubyFT was re-cloned into mammalian vectors pmRubyFT-β-actin and pmRubyFT-α-tubulin, and after transfection of HeLa Kyoto cells, confocal images of the actin filaments and tubulin microtubules were acquired (Figure 5). For mRubyFT-β-actin and mRubyFT-α-tubulin fusions, both blue and red forms were seen as fibers (Figure 5a,b). The red-to-blue ratio images were also calculated and revealed an uneven distribution of actin filaments and tubulin microtubules according to their age-related red-to-blue ratio (Figure 5). Hence, the mRubyFT timer is applicable for the labeling and confocal imaging of the cytoskeleton of the live mammalian cells.

### 2.5. Structural Characterization of mRubyFT Timer

To determine why mRubyFT has the blue-to-red timer properties on the molecular level and to elucidate the impact of mutations introduced during directed molecular evolution, we determined the mRubyFT crystal structure using X-ray at 1.5 Å resolution (Figure 6a). In the crystal, mRubyFT has one protein chain per asymmetric unit, and a contact analysis revealed that the protein is a monomer at pH > 5.5, i.e., in a highly fluorescent red state (Figure 2d). The mRubyFT chromophore formed by ^68^LYG^70^ (corresponding to ^65^LYG^67^ in the alignment, Figure 1) amino acids is positioned on the central helix of the β-barrel (Figure 6a). The structure of mRubyFT demonstrates a clear electron density for a cis-isomer of the red 2-(iminomethyl)-4- (4-hydroxybenzylidene)-imidazol-5-one chromophore with coplanar geometry of imidazolic and phenolic rings (Figure 6b). The chromophore is stabilized by four direct hydrogen bonds (H-bonds) to R72, W95, R97, and S148, and five water-mediated H-bonds to S66, Q111, T113, E146, and L204 (Figure 6c). The phenolic hydroxyl group of the chromophore forms hydrogen bonds with the hydroxyl group of S148 and with the main chain of E146 and L204 via buried water molecule (Figure 6c). The negative charge of the phenolic hydroxyl group of the chromophore is maintained by H202 that is stacked with the phenolic group of the chromophore.

To understand the appearance of the blue form in mRubyFT, we analyzed the mutations in mRubyFT compared to mRuby2 with a focus on the chromophore and its environment. There are four mutations in mRubyFT compared to mRuby2 that are inner to the β-barrel: M68L, N148S, Q218L, and A222S (corresponding to M65L, N148S, Q220L, and A224S in the alignment, Figure 1). All of them were either in the chromophore or in contact with the chromophore, or located at distances in the range of 6 Å away from the chromophore. Leu68 is the first amino acid of the chromophore in mRubyFT. It is known that Leu in this position stabilizes the blue form of the chromophore. mTagBFP [6] and mTagBFP2 [13] bright blue fluorescent proteins contained the LYG chromophore. Therefore, mRubyFT has a bright blue form due to the M68L mutation. The Q218L mutation also seems to be favorable for the stabilization of the blue form of mRubyFT, as L218 together with M46, F67, and L204 formed a hydrophobic environment for L68 moiety of the chromophore (Figure 6c).

To elucidate the effect of mutations introduced during mRubyFT development, we compared the chromophores and their environments for mRubyFT and the parental protein mRuby (PDB ID: 3U0M) [9]. The chromophores in both proteins have planar geometry (Figure 7a). Unlike the cis-isomer of the red chromophore in mRubyFT, the structure of mRuby demonstrates a trans-isomer of the red chromophore. The N148S mutation in mRubyFT is favorable for the stabilization of the cis-isomer due to the strong hydrogen bond (2.4 Å) of the S148 OH-group with the phenolic hydroxyl group of the chromophore (Figure 6c). In addition, this hydroxyl group of the chromophore forms water-mediated hydrogen bonds to E146 and L204, stabilizing the cis-isomer. In mRuby, the phenolic hydroxyl group of the MYG chromophore forms two H-bonds with the side chains of N143 and T158 (S148 and T163 in mRubyFT) (Figure 7b). Note that in both mRuby and mRubyFT structures, the side chain of threonine T158 (T163) has two conformations, regardless of whether it forms a hydrogen bond to the OH-group of the chromophore (Figure 7b).

Compared to the mRuby structure, the A222S mutation in mRubyFT resulted in the formation of two additional H-bonds of S222 with H202 and E220 (corresponding to E222 in the alignment, Figure 1, and E222 in EGFP). The former bond stabilizes the conformation of H202 imidazolic ring that forms a stacking interaction with the tyrosine moiety of the chromophore. However, compared to the mRuby structure, where H202 and chromophore imidazolic moiety are almost coplanar, the S222–H202 hydrogen bond in mRubyFT stabilizes the side chain of H202 in a conformation that is less coplanar to the phenolic moiety of the chromophore—rotated about 15° compared to that in mRuby (Appendix A). In addition, A222S substitution leads to the loss of the hydrogen bond between the side chains of E220 and H202 found in mRuby (Figure 7b). This probably leads to the appearance of two alternative conformations of E220 residue with equal occupancies in mRubyFT. Of note, E220 is the key residue for the formation and maturation of the chromophores in GFP-like fluorescent proteins [14].

We further compared the chromophores and their environment for two blue-to-red fluorescent timers: Fast-FT (PDB ID: 3LF3) [4] and mRubyFT. Both FTs contained cis-isomers of the red chromophore (Figure 8). While mRubyFT in the crystal had a totally intact chromophore, the chromophore of Fast-FT underwent an unusual degradation, resulting in the complete elimination of the polypeptide chain of Met66 (Fast-FT numbering), the first chromophore-forming residue, introducing a break in the protein backbone between Phe65 and the chromophore. Among two alternative states of the chromophore in the Fast-FT crystal structure, the degraded form has 0.8 occupancy, while the non-degraded cis-conformation only has 0.2. Therefore, the structural data evidence shows that the chromophore in mRubyFT was more stable over time. In contrast to E220 in mRubyFT, the side chain of the corresponding E215 in Fast-FT is tightly hydrogen-bonded to the N2 nitrogen of the chromophore imidazolic group, but this is possibly connected to the degraded state of the chromophore in Fast-FT, as in the case of the low-occupancy chromophore state, the E215 side chain has too short a distance (1.6Å) to the M66 group of the chromophore. Both FTs contained serine residue in position 222 (217 in Fast-FT), that is directly (mRubyFT) or via a solvent molecule bound to the E220 (E215) side chain (Figure 8b). Interestingly, both mutations in Fast-FT, Fast-FT/S217A, and Fast-FT/S217C decreased the rate of blue and red chromophore formation, but resulted in the formation of the bright chromophores. On the contrary, the S222A (S224A according to the alignment, Figure 1) substitution of mRubyFT led to an acceleration of the maturation of blue and red forms (Table 3), and S222C (S224C according to the alignment, Figure 1) completely blocked the formation of both blue and red chromophores. Thus, S222 is responsible for the timer characteristics and maturation of mRubyFT.

In comparison with mRuby2, mRubyFT also has two mutations, N130D and L229R (N129D and L231R in the alignment, Figure 1), that are outer to the β-barrel. As mRubyFT is a monomer in the crystal, we analyzed the contacts of the corresponding N125 and L224 amino acids in the eqFP611 tetrameric red fluorescent protein (PDB ID: 3E5W) [15] that is parental to mRuby. N125 and L224 are located on the surface of the subunits and engaged in contacts of AC and AD inter-subunit interfaces. N125 from the A-subunit is located near the negatively charged E91 and E175 amino acids from the C-subunit. The hydrophobic side chain of the L224 from the A-subunit fits in a small volume within the AD interface, where it coexists with a side chain of V218 from the adjacent subunit. In addition, L224 from the A-subunit is located near the positively charged R198 and H216 from the D-subunit. Therefore, the introduction into mRubyFT of the negatively charged D130 and positively charged R229 with a large side chain instead of N130 (N125 in eqFP611) and L229 (L224 in eqfp611), respectively, must stabilize the monomeric state and perturb potential AC and AD interfaces.

In summary, we speculate that mRubyFT has a bright blue form due to M68L and Q218L mutations (corresponding to M65L and Q220L in the alignment, Figure 1). The A222S mutation (corresponding to A224S in the alignment, Figure 1) is responsible for the timer characteristics and maturation of both forms in mRubyFT. In addition, based on the crystal structure data, the chromophore in mRubyFT lacks any signs of degradation and has a single cis-configuration stabilized by the N148S mutation.

### 2.6. Directed Mutagenesis of the mRubyFT Blue-to-Red Fluorescent Timer and mRuby2 RFP at Key Positions

We suggested that the residues in positions 65 and 220 may be the key residues for the appearance of the blue-to-red timer-like phenotype in the mRuby2 protein. However, when expressed in bacterial cells, mRuby2/M65L, mRuby2/Q220L, and double mRuby2/M65L/Q220L mutants revealed dim red fluorescence, but with no signs of blue fluorescence.

Next, we carried out directed mutagenesis of the mRubyFT timer at the amino acid residues 62, 69, 148, 165, 167, 203, and 224, located in the immediate environment of the chromophore according to the structural data, and characterized the characteristic maturation times for the blue and red forms (Table 3). The R69K and S224C mutants of mRubyFT were non-fluorescent; in contrast, the introduction of mutations R70K, S217A, and S217C into the Fast-FT timer at analogous positions did not affect the efficient formation of the blue and red forms [4]. Substitutions in mRubyFT at positions 148, 165, and 167 led to the disappearance of the fluorescence of the red form; however, the fluorescence of the blue form was retained (Table 3). In the case of the S148I substitution, the blue fluorescence was stable over time (thus leading to the formation of a blue variant resembling the mTagBFP-like protein); mutation S146A in Fast-FT at an analogous position also resulted in the stabilization of the blue form and blocked the formation of the red form [4]. The T62S mutation accelerated the maturation of the red form and had practically no effect on the maximum time for the blue form (Table 3). The double mutation R69K/H203Y (identical to the residues in the same positions in the blue mTagBFP protein) led to an acceleration of the maturation of both the blue and red forms 34- and 14-fold, respectively (thus, this double mutant was an example of a very fast blue-to-red timer). The H203Y mutation in the mRubyFT timer led to a 15-fold acceleration of the maturation of the red form and the formation of a stable blue form (Table 3). S224A substitution led to an acceleration of the maturation of blue and red forms by 4- and 27-fold, respectively (Table 3). Thus, residues 62, 69, 148, 165, 167, 203, and 224 from the immediate environment of the chromophore are important for the formation and maturation rate of the blue and red forms of the mRubyFT timer.

## 3. Materials and Methods

### 3.1. Cloning of Bacterial Vectors, Mutagenesis and Library Screening

mRuby2, mRubyFT, and its variants were cloned into the pBAD/HisB plasmid (Invitrogen, Waltham, MA, USA) at the BglII/EcoRI restriction sites using the Fw-BglII-(PA)TagRFP/Rv-GFP-EcoRI primers listed in Appendix A to express these proteins in BW25113 bacterial cells (kindly provided by Verkhusha V.V. from Albert Einstein College of Medicine, Bronx, NY, USA).

Random libraries for the development of mRubyFT were obtained using polymerase chain reaction (PCR) in the presence of Mn^2+^ ions in the conditions of 2–3 random mutations per 1000 base pairs (Diversify PCR Random Mutagenesis Kit User Manual, Clontech, Palo Alto, CA, USA) and cloned at the BglII/EcoRI restriction sites of the pBAD/HisB plasmid. For PCR, the C1000 Touch Thermal Cycler (Bio-Rad, Hercules, CA, USA) was used.

An overlap library for the rational mutagenesis of the parental mRuby2 protein at positions 65, 148, 165, 167, 220, and 224 was generated using the mRubyFT-65, mRubyFT-65-r, mRubyFT-148, mRubyFT-148-r, mRubyFT-165, mRubyFT-165-r, and mRubyFT-220-r primers listed in Appendix A. The assembly of the whole genes was performed using PCR with overlapping fragments [16]. The generated library was inserted at the BglII/EcoRI restriction sites of the pBAD/HisB plasmid.

Directed mutagenesis of the mRubyFT protein at positions 62, 69, 148, 165, 167, 203, and 224 and the mRuby protein at positions 65 and 220 was performed using the mRubyFT-X/mRubyFT-X-r and mRuby-X/mRuby-X-r corresponding primers listed in Appendix A. The assembly of the gene was performed using PCR with overlapping fragments [1]. The resulting genes were inserted at the BglII/EcoRI restriction sites into the pBAD/HisB plasmid.

The resulting PCR products were purified using a DNA purification kit (Eurogen, Russia). After the digestion of the pBAD/HisB plasmid and PCR products with the corresponding restriction enzymes, they were purified on a 1% agarose gel, followed by extraction using a DNA extraction gel kit (Eurogen, Moscow, Russia). After extraction, the plasmids and PCR products were ligated by T4 DNA ligase (3–16 h, at 16 °C), followed by purification of the ligated mixture with a DNA purification kit (Eurogen, Moscow, Russia). The purified ligation mixture was further transformed into electrocompetent bacteria BW25113 by electroporation using 1800 V pulse in an Eporate electroporator (Eppendorf, Hamburg, Germany).

The screening of the bacterial libraries was performed on Petri dishes under a fluorescent microscope. Briefly, the expression of the timers on the colonies on Petri dishes was induced with 0.2% arabinose at 37 °C. The screening of about 10,000 colonies of the bacterial library expressing FT variants was performed on Petri dishes under fluorescent stereomicroscope Leica M205FA (Leica, Wetzlar, Germany) equipped with the DFC310FX camera (Leica Microsystems, Wetzlar, Germany) and a mercury metal halide light source EL6000 (Leica Microsystems, Wetzlar, Germany). Blue fluorescence was registered by 405/40BP excitation and 450/40BP emission filters. Red fluorescence was registered by 540/40BP excitation and 620/60BP emission filters. We marked the bluest/non-red colonies 18 h after plating the bacteria on Petri dishes and selected the most red/non-blue colonies 72 h after plating. The acquired images were analyzed using ImageJ software, and the colonies with the largest contrast and brightness of both forms were selected for further analysis on bacterial streaks on Petri dishes, followed by protein purification and characterization.

After each round, 10–15 selected clones were purified and their properties characterized. To assess the red form, the FT mutants were expressed in 10 mL of LB medium (10 g tryptone, 5 g yeast extract, and 10 g NaCl per 1L of water) containing ampicillin (100 µg/mL) and 0.002% arabinose for 16 h at 37 °C, 220 rpm and for 24 h at room temperature, followed by purification on Ni-NTA resin as described below. To characterize the blue form, the mutants were expressed in 100 mL of LB medium supplemented with ampicillin (100 µg/mL) in the absence of arabinose at 37 °C, 190 rpm, overnight. Then, 0.2% arabinose was added and the bacterial culture was incubated for 2–4 h at 37 °C, 190 rpm with a restriction of oxygen in 1L flasks closed with parafilm. The bacteria were precipitated by centrifugation for 12 min at 3500 rpm. The proteins were further extracted from the pellet with 300 μL of B-Per extraction reagent containing lysozyme (1 mg/mL final concentration) and DNase I (1 unit/μL) by shaking the pellet for 20 min at 37 °C and 200 rpm. Next, the lysed components of the bacteria were removed by centrifugation for 2 min at 13,200 rpm. The protein supernatant was further bound to 150 μL of Ni-NTA resin for 30–60 min on an orbital shaker at 4 °C. Next, the resin with bound protein was washed twice with 1 mL of a PBS buffer and once with 1 mL of 10 mM imidazole (pH 8.0) in a PBS buffer. The protein was then eluted from a column packed with resin with 400 mM imidazole in the PBS buffer. Finally, using a spectrofluorometer and spectrophotometer, the characteristic maturation times and brightness were determined for the purified proteins, as described in Section 3.2.

### 3.2. Proteins’ Purification and Characterization

For the final characterization, the proteins were expressed and purified using the pBAD/HisB arabinose-inducible system (Invitrogen, Waltham, MA, USA) from 400 mL of medium, as described in Section 3.1 with modifications. Briefly, for red-form protein expression, the bacterial cultures were grown in 400 mL of LB medium supplemented with 0.004% arabinose and 100 μg/mL ampicillin overnight at 37 °C and 220 rpm. For blue-form protein expression, the bacterial cultures were grown in 400 mL of LB medium supplemented with 100 μg/mL ampicillin overnight at 37 °C and 220 rpm; the next day, the protein expression was induced with 0.2% arabinose for 4 h at 37 °C and 220 rpm. The cultures were then centrifuged at 4648× *g* for 10 min. The cell pellets were resuspended in PBS buffer, pH 7.4, supplemented with 300 mM NaCl (buffer A) and 10 mM imidazole, and lysed by sonication on ice (for 8 min, in the cycle of 30 sec pulse, and 30 sec pause; 20% power of the VCX130 Sonicator and CV18 tip (Sonics & Materials Inc., Newtown, CT, USA). The sonicated solution was centrifuged at 36,670× *g* at 4 °C for 4 min. The proteins were further bound with 1–1.5 mL of Ni-NTA resin (Qiagen, Germantown, MD, USA) for 30 min on ice with mixing. Resin with bound protein was twice washed with buffer A. The proteins were eluted with 400 mM imidazole in buffer A. The collected fractions with eluted protein of 1–1.5 mL were dialyzed against PBS buffer for 16 h.

The extinction coefficient values for the blue form of purified mRubyFT protein and its derivatives were calculated in PBS buffer, pH 7.4, using the acid denaturation method and assuming that the TagBFP-like chromophore has the extinction coefficient of 28,500 M^−1^ cm^−1^ at 382 nm in 1M HCl [1]. The extinction coefficient values for the red form of purified mRubyFT and Fast-FT proteins were calculated in PBS buffer, pH 7.4, relative to the absorption peak at 280 nm, assuming the extinction coefficient at 280 nm of 26,025 and 39,880 M^−1^ cm^−1^, respectively. The absorption spectra were recorded using a NanoDrop 2000c Spectrophotometer (Thermo Scientific, Waltham, MA, USA).

The quantum yields for the blue form of the purified mRubyFT protein and its derivatives excited at 400 nm were measured by a comparison of the integrated fluorescence values (in the range of 410–800 nm) in PBS buffer, pH 7.40, with the similarly integrated fluorescence values for the equally absorbing at 400 nm mTagBFP2 protein (quantum yield of 0.64 [13]). The quantum yields for the red form of the purified mRubyFT protein and its derivatives excited at 540 nm were measured by a comparison of the integrated fluorescence values (in the range of 550–820 nm) in PBS buffer, pH 7.40, with the similarly integrated fluorescence values for the equally absorbing at 540 nm mCherry protein (quantum yield of 0.22 [17]). The fluorescence spectra were acquired using a CM2203 spectrofluorometer (SOLAR, Minsk, Belarus).

The pH titrations for the purified mRubyFT protein (1.2 µM final concentration) were performed in a buffer of 30 mM citric acid, 30 mM borax, and 30 mM NaCl with a pH adjusted from 3.0 to 10.5, after incubation for 20 min at room temperature. Blue (Ex 365 nm/Em 410–460 nm) and red fluorescence (Ex 525 nm/Em 580–640 nm) was registered using a 96-well Modulus^TM^ II Microplate Reader (Turner Biosystems, Sunnyvale, CA, USA).

Size-exclusion chromatography was performed with a Superdex^TM^ 75 10/300 GL column using the GE AKTA Explorer 100 (Amersham Pharmacia, UK) FPLC System.

To assess the maturation rate of mRubyFT and its derivatives, 100 mL of bacterial cultures were grown in a 1 L flask with LB medium supplemented with 100 µg/mL ampicillin at 37 °C, 190 rpm, overnight. Next, protein expression was induced by the addition of 0.2% arabinose, and the flask throat was closed using parafilm. The protein expression lasted for 2–4 h at 37 °C, 190 rpm, under anaerobic conditions. The cultures were then centrifuged at 3500× *g* for 12 min at room temperature. The protein was purified on ice using Ni-NTA resin. A total of 100 µL of purified protein was mixed with 2.9 mL of PBS buffer supplemented (pre-warmed at 37 °C for 10 min) in a 5 mL quartz cuvette. Fluorescence kinetics were further measured using the CM2203 spectrofluorometer (SOLAR, Minsk, Belarus) at 37 °C with registration of both blue (Ex 400 nm/Em 460 nm) and red fluorescence (Ex 580 nm/Em 630 nm) changes over time.

For the preparative purification of the mRubyFT protein for X-ray crystallography, bacterial cells expressing the mRubyFT protein with N-terminal His-tag and the Tobacco Etch Virus (TEV) protease cleavage site were pelleted by centrifugation for 20 min at 5000 rpm and 4 °C (Beckman Coulter centrifuge, Brea, CA, USA). Then, the pellet (pellet weight was 14 g from 2.6 L of medium) was resuspended in 100 mL of buffer A (40 mM Tris-HCl, pH 7.8, containing 400 mM NaCl and 10 mM imidazole) supplemented with 0.2% Triton X-100, and 1 mM phenylmethylsulfonyl fluoride, and disrupted by ultrasound sonication (2 s pulse, 6 s pause, amplitude 45%, total time 5 min). The crude cell extract was centrifuged for 30 min at 28,000× *g* and 4 °C (Beckman Coulter centrifuge, Brea, CA, USA). The supernatant was loaded onto a 5 mL Ni-NTA Superflow column (Qiagen, Hilden, Germany) equilibrated with buffer A supplemented with 0.1% (*v*/*v*) Triton X-100. Then, there were sequential washes with buffer A and buffer A supplemented with 40 mM imidazole. Protein elution was performed with buffer A supplemented with 300 mM imidazole (Appendix A, elution). Amounts of 1 mM DTT and 1 mM EDTA were added to the protein solution, mixed with TEV protease (1 mg per 10 mg protein), and the whole mixture was dialyzed for 16 h in buffer B (40 mM Tris, pH 7.8, 400 mM NaCl, 5 mM imidazole, 2 mM BME, 1 mM EDTA) at +4 °C; His-tag cleavage was controlled by electrophoresis in polyacrylamide gel (PAGE), gel concentration 15% (Appendix A, proteolysis). The digested protein was then loaded onto a Ni-NTA Superflow column (Qiagen, EU) equilibrated with buffer B; TEV protease and cleaved His-tag bound to a Ni-NTA Superflow column (Qiagen, EU), and the leaked protein was concentrated to 2.2 mL using a 10 kDa cut-off concentrator (Millipore, Burlington, MA, USA) and loaded onto a Superdex column 75 10/300 GL (GE Healthcare, Danderyd, Sweden) in buffer 20 mM Tris-HCl pH 7.8, 150 mM NaCl. The protein was eluted in a volume of 12.3–12.9 mL with a maximum peak of 12.7 mL in absorbance at 280 nm (Figure 2e). Fractions containing pure protein were concentrated till 10 mg/mL concentration using 10 kDa cut-off concentrators (Millipore, Burlington, MA, USA). Then, the protein concentration was measured by the bicinchoninic method using the Bicinchoninic Acid Protein Assay Kit (Sigma-Aldrich, Saint Louis, MO, USA). BSA protein standard P0914-5AMP solution (Sigma-Aldrich, Saint Louis, MO, USA) was used as a standard. The total protein yield was 18 mg. The purity of the preparations at all stages was monitored by electrophoresis in PAGE (gel concentration 15%). Protein chromatography was performed using the ÄKTA prime plus and ÄKTA explorer 100 systems (GE Healthcare, Danderyd, Sweden).

### 3.3. Protein Crystallization

An initial crystallization screening of mRubyFT was performed with a robotic crystallization system (Rigaku, Woodlands, TX, USA) and commercially available 96-well crystallization screens (Hampton Research, Aliso Viejo, CA, USA and Anatrace, Maumee, OH, USA) at 15 °C using the sitting drop vapor diffusion method. The protein concentration was 15 mg/mL in the following buffer: 20 mM Tris-HCl, 200 mM NaCl, pH 7.8. Optimization of the initial conditions was performed by the hanging-drop vapor-diffusion method in 24-well VDX plates. Crystals were obtained within 2 weeks in the following conditions: 0.1M bis-tris pH 5.5; 21% PEG 3350.

### 3.4. Data Collection, Processing, Structure Solution, and Refinement

mRubyFT crystals were briefly soaked in a 100% Paratone oil (Hampton Research, Aliso Viejo, CA, USA) immediately prior to diffraction data collection and flash-frozen in liquid nitrogen. The crystals were preliminarily tested for their diffraction quality at the beamline “Belok-RSA” of the Kurchatov SNC (Moscow, Russia) [18]. The high-resolution data at 1.5Å were collected at 100K at BL41XU beamline (SPring8, Sayo, Hyogo, Japan). The data were indexed, integrated, and scaled using the DIALS program [19] (Table 4). The program Pointless [20] suggested the P2_1_2_1_2_1_ space group.

The structure was solved by the molecular replacement method using the MOLREP program [21] and the structure of the fluorescent protein mRuby (PDB ID: 3U0L) as an initial model. The refinement of the structure was carried out using the REFMAC5 program of the CCP4 suite [22]. The visual inspection of electron density maps and the manual rebuilding of the model were carried out using the COOT interactive graphics program [23]. The resolution was successively increased to 1.50 Å, and the hydrogen atoms in fixed positions, as well as TLS were introduced during the final refinement cycles. In the final model, an asymmetric unit contained one copy of the protein of 220 residues with the chromophore, 158 water molecules, and three magnesium ions from the protein buffer. The first seven residues from the N-terminal as well as the last ten residues from the C-terminal part of the protein were not visible in the electron density due to disorder.

### 3.5. Structure Analysis and Validation

The visual inspection of the structure was carried out using the COOT program and the PyMOL Molecular Graphics System, Version 1.9.0.0 (Schrödinger, New York, NY, USA). The structure comparison and superposition were made using the PDBeFold program [24], while contacts were analyzed using the PDBePISA [25] and WHATIF software [26]. The chromophore environment was visualized by LigPlot+ v.2.2.4 [27].

### 3.6. Mammalian Plasmids Construction

In order to construct the pAAV-*CAG*-mRubyFT-P2A-EGFP and pAAV-*CAG*-Fast-FT-P2A-EGFP plasmids, the mRubyFT and Fast-FT genes were PCR amplified as the KpnI-AgeI fragments, using RubyFT-NheI2/RubyFT-AgeI-r or mCherry-KpnI/RubyFT-AgeI-r primers listed in Appendix A, and swapped with the iRFP gene in the pAAV-*CAG*-iRFP-P2A-EGFP vector.

In order to construct the pmRubyFT-actin plasmid, the mRubyFT gene was PCR amplified as the NheI-BglII fragment, using RubyFT-NheI2/RubyFT-BglII-r primers listed in Appendix A, and swapped with the TagBFP gene in the pTagBFP-actin vector (Evrogen, Moscow, Russia).

In order to construct the pmRubyFT-tubulin plasmid, the mRubyFT gene was PCR amplified as the NheI-BglII fragment, using RubyFT-NheI2/RubyFT-BglII-r primers listed in Appendix A, and swapped with the TagGFP2 gene in the pTagGFP2-tubulin vector (Evrogen, Moscow, Russia).

In order to construct the pTRE-mRubyFT plasmid, the mRubyFT gene was PCR amplified as the EcoRI-XbaI fragment, using RubyFT-EcoRI/LSSmSc-XbaI-r primers listed in the Appendix A, and inserted in pTRE vector at EcoRI/XbaI restriction sites.

### 3.7. Mammalian Live Cell Imaging

Mammalian live cell imaging was performed as described earlier [28]. Briefly, transient transfection of the HeLa Kyoto cells was performed in a 24-well format using lipofectamine reagent according to the manufacturer’s protocol. Cells were cultured using DMEM medium supplemented with 10% FBS, glutamine, 50 U/mL penicillin, and 50 U/mL streptomycin, at 37 °C and 5% CO_2_. HeLa cell cultures were imaged 24–72 h after the transient transfection using a laser spinning-disk Andor XDi Technology Revolution multi-point confocal system (Andor Technology, Belfast, UK) equipped with an inverted Nikon Eclipse Ti-E/B microscope (Nikon Instruments, Tokyo, Japan), a 75 W mercury–xenon lamp (Hamamatsu, Hamamatsu, Japan), a 60× oil immersion objective NA 1.4 (Nikon, Tokyo, Japan), a 16-bit Neo sCMOS camera (Andor Technology, Belfast, UK), a laser module Revolution 600 (Andor Technology, Belfast, UK), and a spinning-disk module Yokogawa CSU-W1 (Andor Technology, Belfast, UK). The blue, green, and red fluorescence were acquired using the 405, 488, and 561 nm lasers, a confocal dichroic mirror 405/488/561/640, and filter wheel emission filters 447/60, 525/50, and 617/73, respectively. During imaging, the cells were incubated at 37 °C and 5% CO_2_ using a cage incubator (Okolab, Naples, Italy).

### 3.8. Statistics

To estimate the significance of the difference between two values, we used the Mann–Whitney Rank Sum Test and provided *p*-values (throughout the text in the brackets) calculated for the two-tailed hypothesis. We considered the difference significant if the *p* value was <0.05.

## 4. Conclusions

In conclusion, we developed and characterized the novel blue-to-red fluorescent timer derived from mRuby2 RFP.

Earlier, we suggested that any RFPs with the tyrosine-based chromophore most likely can be converted into blue variants by the introduction of 65L,H/69K/84F,W,L/148F,I/165A,I,N/181A,I/203F,Y mutations [6]. In this study, we were successful in the conversion of the other RFP with the tyrosine-based chromophore (mRuby2) into the stable blue probe, called mRubyFT/S148I (Table 3). The mRubyFT/S148I blue mutant contained M65L, N148I, Q220L, and A224S internal mutations relative to the original mRuby2 RFP and 69R, 84F, 165T, 181F, and 203H residues in the positions, suggested earlier for the generation of the blue variant. Hence, we can expand the rule suggested earlier to the following phrase, that any RFPs with the tyrosine-based chromophore can most likely be converted into a blue probe by the introduction of 65L,H/69K,R/84F,W,L/148F,I/165A,I,N,T/181A,I,F/203F,Y,H/220L/224S mutations.

We speculate that the chemical structure of the chromophore in the blue fluorescent form of the mRubyFT protein can be identical to the chemical structure of the mTagBFP protein [8]. This assumption is supported by the fact that the blue forms of the mRubyFT and mTagBFP proteins share the same LYG-chromophore tripeptide and have similar absorption/emission maxima peaked at 406/457 (Table 1) and 400/455 nm [8], respectively. However, further investigation is necessary to prove this assumption.

In addition to the low brightness of FTs, the practical application of the blue-to-red FTs is limited by two obstacles. First, blue-to-red transition occurs even at a storage of 4 °C (Appendix A). This limitation can be overcome by the storage in anaerobic conditions, e.g., in the atmosphere of carbon dioxide. The second limitation of the blue-to-red FTs is related with violet light-induced transformation of the blue form into red, resulting in the appearance of the red form during imaging of the blue form [1]. For mRubyFT, the efficiency of the blue-to-red photoconversion with 395 nm light was 4.2-fold less than for Fast-FT timer (Figure 3c–e). This undesired blue-to-red photoconversion can be avoided by the imaging of the red form first and the blue form afterward.

## Figures and Tables

**Figure 1 ijms-23-03208-f001:**
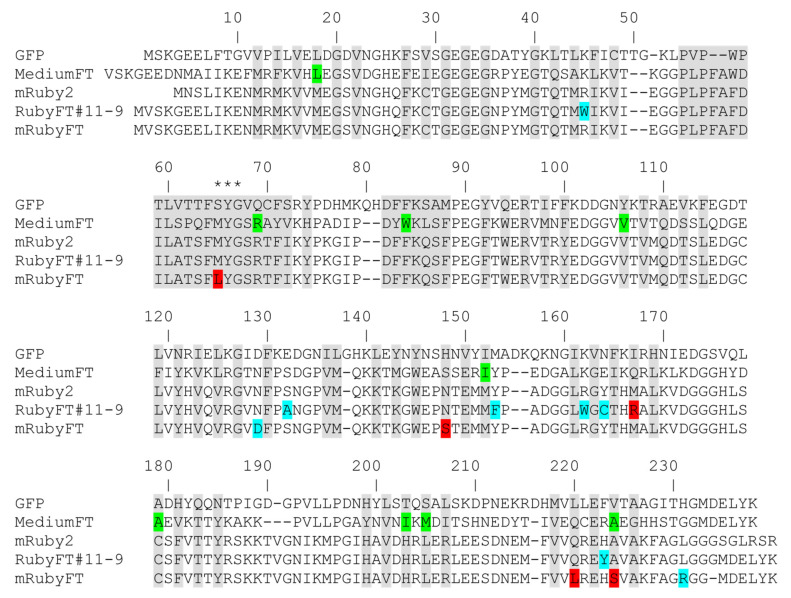
Amino acid sequence alignment for RubyFT and Medium-FT timers and GFP and mRuby2 fluorescent proteins. Alignment numbering follows that of Aequorea victoria GFP. The residues inside the β-barrel are highlighted in gray. Asterisks indicate three residues that form the chromophore tripeptide. Internal and external mutations in RubyFT timers relative to the mRuby2 original matrix are highlighted in red and cyan colors, respectively. In the case of Medium-FT, internal and external mutations relative to the mCherry protein are highlighted in green.

**Figure 2 ijms-23-03208-f002:**
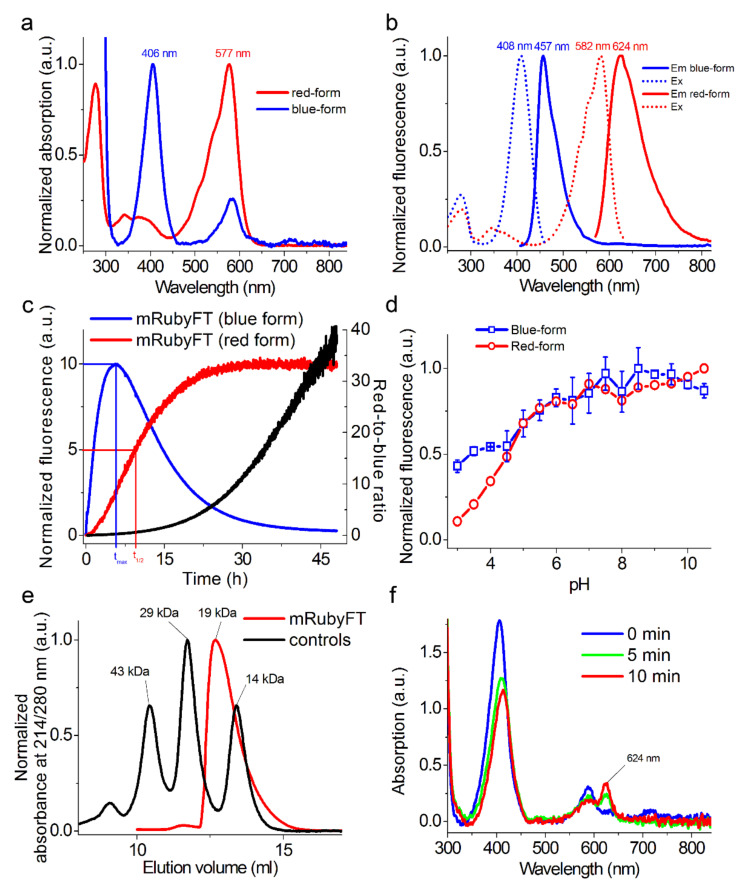
In vitro properties of the purified mRubyFT protein. (**a**) Absorption spectra for blue and red forms of mRubyFT protein in PBS buffer at pH 7.40. (**b**) Excitation and emission spectra for blue and red forms of mRubyFT in PBS buffer at pH 7.40. (**c**) Maturation of blue and red forms for mRubyFT in PBS buffer at pH 7.40, 37 °C. Red-to-blue ratio was calculated according to the red and blue fluorescence time dependences normalized to 100. (**d**) Fluorescence intensity for blue and red forms of mRubyFT as a function of pH. Three replicates were averaged for analysis. Error bars represent the standard deviation. (**e**) Fast protein liquid chromatography of mRubyFT protein. mRubyFT was eluted in 20 mM Tris-HCl (pH 7.80) and 200 mM NaCl buffer. The molecular weight of mRubyFT was calculated from a linear regression of the dependence of logarithm of control molecular weights vs. elution volume (Appendix A). (**f**) Changes in absorption spectrum of the purified mRubyFT in PBS buffer, pH 7.4 as a result of illumination with 405 nm LED array (57 mW/cm^2^) for the indicated time.

**Figure 3 ijms-23-03208-f003:**
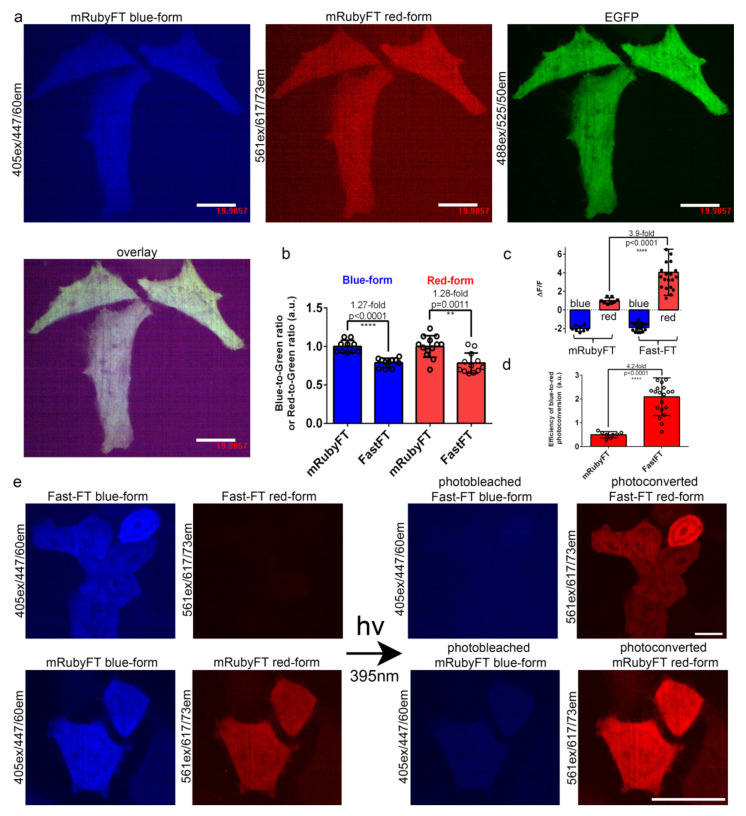
Comparison of the brightness and blue-to-red photoconversion of mRubyFT and control Fast-FT timers in live HeLa cells. (**a**) Confocal images of live HeLa cells expressing mRubyFT-P2A-EGFP fusion. P2A is a self-cleavable peptide. Blue (405ex and 447/60em), green (488ex and 525/50em), and red (561ex and 617/73em) fluorescence channels and their superposition are shown. (**b**) The averaged brightness for the blue and red forms for the mRubyFT and Fast-FT timers in HeLa cells normalized to the brightness of the EGFP protein expressing in the same cell. Error bars are standard deviations across 10–13 cells. (**c**) The mean ΔF/F values for photobleached blue form and photoconverted red form of mRubyFT and control Fast-FT timers expressed in live HeLa cells 24 h after transfection. The pulse of 395/25 nm light (0.9 mW/cm^2^ power measured before 60x oil objective lens) lasted for 1 min. Error bars are standard deviations across 8–20 cells. (**d**) Efficiency of the blue-to-red photoconversion with 395/25 nm light during 1 min was calculated as a ΔF/F_red_/ΔF/F_blue_. (**e**) Confocal images of live HeLa cells expressing mRubyFT-P2A-EGFP or Fast-FT-P2A-EGFP fusion. Blue (405ex and 447/60em) and red (561ex and 617/73em) fluorescence channels before and after irradiation with 395/25 nm light of 1 min duration are shown. Protein expression lasted 72 h. For red and blue images, the contrast settings were the same. Images were acquired 72 h (**a,b**) or 24 h (**c**–**e**) after transfection. (**b**–**d**) *p* values show statistical difference between the respective values. ****, *p* value is < 0.0001. **, *p* value is < 0.01. (**a,e**) Scale bars: 20 µm.

**Figure 4 ijms-23-03208-f004:**
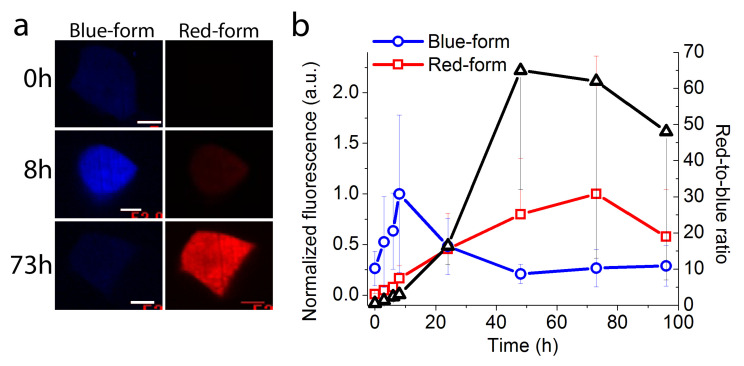
Timer behavior of the mRubyFT protein transiently expressed in live HEK293T cells under the control of the Tet-Off system. (**a**) Confocal images of the live HEK293T cells 0, 8, and 73 h after the doxycycline addition (2 µg/mL final) in blue (405ex and 447/60em) and red (561ex and 617/73em) fluorescence channels. For different time points, the image contrast settings were the same. Doxycycline was added 16 h after co-transfection of HEK293T cells with pTRE-mRubyFT, pTET off, and pLU-CMV-GFP mix of plasmids (800:800:400 ng, respectively) in 24 well plate. Scale bar: 52 µm. (**b**) Time-dependence of the blue (blue line) and red (red line) fluorescence of the mRubyFT protein and red-to-blue ratio (black line) averaged across 8–16 cells. Error bars are standard deviations.

**Figure 5 ijms-23-03208-f005:**
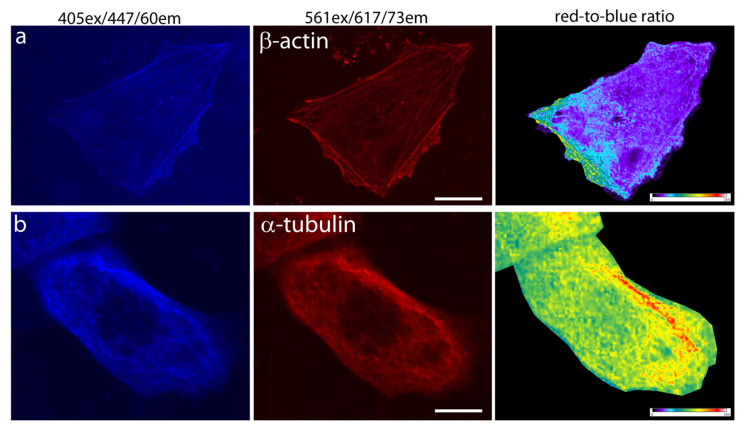
Confocal imaging of the mRubyFT timer in fusions with cytoskeleton proteins in live mammalian cells. Confocal images of HeLa Kyoto cells in blue (405ex and 447/60em) and red (561ex and 617/73em) channels and red-to-blue ratio 72 h after transfection with plasmid (**a**) pmRubyFT-β-actin and (**b**) pmRubyFT-α-tubulin. Scale bar: 10 µm. For blue-to-red ratio image calculation, the background was subtracted and ratio image was generated using ImageJ software; the background ratio was manually cut around the cells.

**Figure 6 ijms-23-03208-f006:**
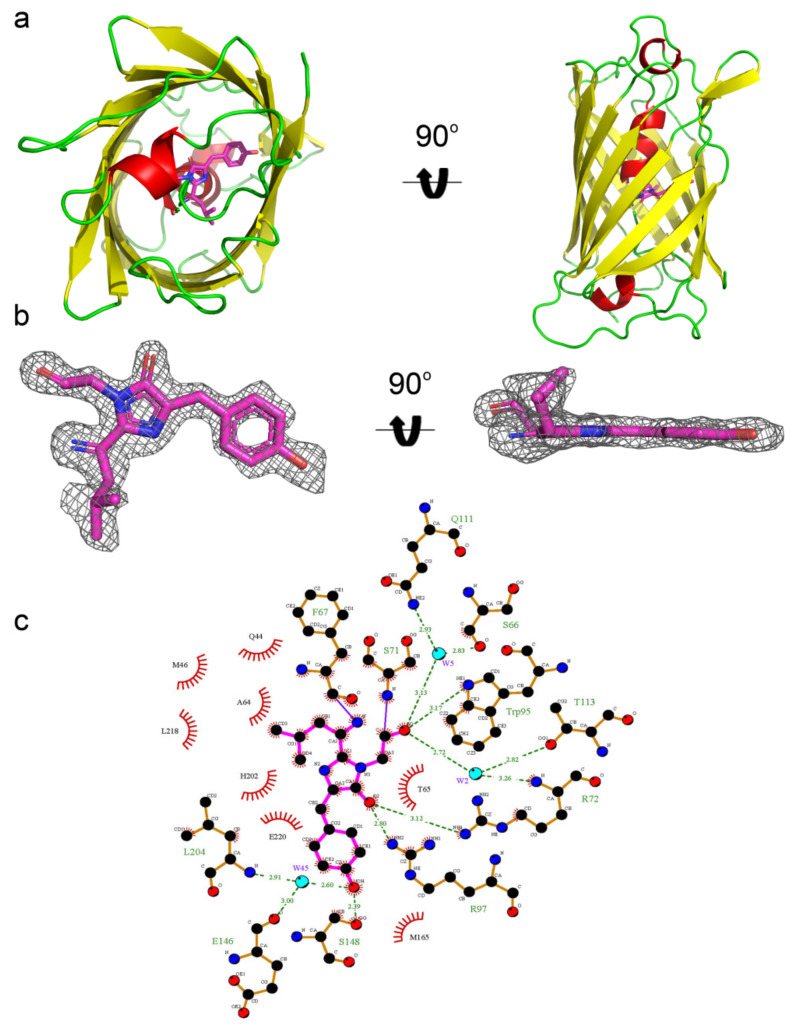
X-ray structure of the red form of the mRubyFT protein. (**a**) Cartoon representation of the overall red mRubyFT monomer. Chromophore, β-sheets, α-helixes, and loop regions are shown in pink, yellow, red, and green colors, respectively. The orientation of the panel on the right is rotated 90° around the horizontal axis with respect to that on the left. (**b**) The omit electron density map (Fo-Fc) around the chromophore of the red mRubyFT protein. The map is contoured at 1.0σ level and shown as gray mesh. The orientation of the chromophore on the right is rotated 90° around the horizontal axis with respect to that on the left. (**c**) The immediate environment of the red mRubyFT chromophore. Chromophore is shown in magenta, residues forming hydrogen bonds in orange, and other residues nearby are shown in red. Hydrogen bonds are depicted as green dashed lines and correspondent distances are labelled. Water molecules are shown as cyan spheres.

**Figure 7 ijms-23-03208-f007:**
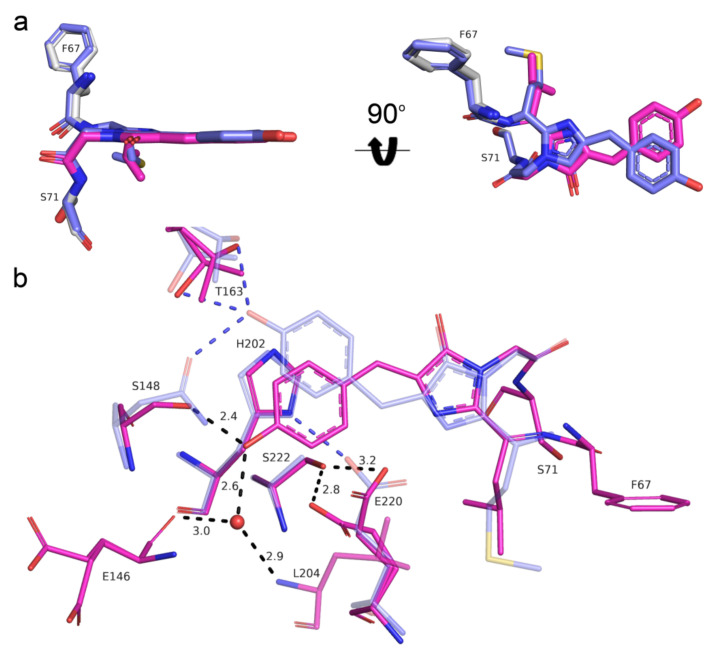
Structural comparison of the chromophores (**a**) and their immediate environments (**b**) for the red form of the mRubyFT timer (pink) and mRuby (PDB ID: 3U0M, blue) protein. Water molecule (red sphere) and hydrogen bonds (dashed lines) are shown. Hydrogen bond distances are only labelled for mRubyFT for clarity. Residues’ enumeration is shown for the mRubyFT protein. In panel (**a**), the orientation of the chromophore on the right is rotated 90° around the horizontal axis with respect to that on the left.

**Figure 8 ijms-23-03208-f008:**
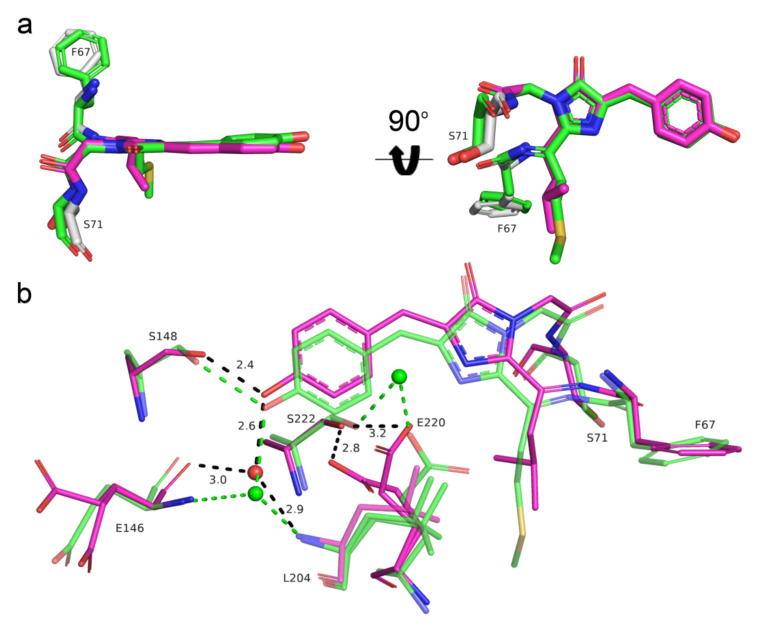
Structural comparison of the chromophores (**a**) and their immediate environments (**b**) for the red form of the mRubyFT (magenta) and red form of the Fast-FT timer (PDB ID: 3LF3, green). Water molecules (red or green spheres) and hydrogen bonds (dashed lines) are shown. Hydrogen bond distances are only labelled for mRubyFT for clarity. Residues’ enumeration is shown for the mRubyFT protein. In panel (**a**), the orientation of the chromophore on the right is rotated 90° around the horizontal axis with respect to that on the left.

**Table 2 ijms-23-03208-t002:** Comparison of the brightness of the mRubyFT timer derived from mRuby2 protein and the mCherry-based Fast-FT timer transiently expressed in HeLa mammalian cells. The blue and red font color reflects the fluorescence color of the respective blue and red form.

Timer	Form	Brightness vs. EGFP (%)	Brightness vs. Fast-FT (%)
mRubyFT	Blue	10.7 ± 0.9	127
	Red	10 ± 2	128
Fast-FT	Blue	8.4 ± 0.7	100
	Red	8 ± 2	100

**Table 3 ijms-23-03208-t003:** In vitro characteristic times for blue and red forms of the purified mutants of mRubyFT timer. ^a^ corresponds to half-time, not maximum, because fluorescence of the blue form reached a plateau.

Protein	Characteristic Times (h)	
	Blue	Red
mRubyFT	5.7	15
mRubyFT/T62S	6.3	0.57
mRubyFT/R69K	Non-fluorescent	Non-fluorescent
mRubyFT/R69K/H203Y	0.17	0.7
mRubyFT/S148F	1.1	Non-fluorescent
mRubyFT/S148I	(5.8) ^a^	Non-fluorescent
mRubyFT/T165N	3.4	Non-fluorescent
mRubyFT/167Q	7.2	Non-fluorescent
mRubyFT/H203Y	(0.72) ^a^	0.65
mRubyFT/S224C	Non-fluorescent	Non-fluorescent
mRubyFT/S224A	1.6	0.35

**Table 4 ijms-23-03208-t004:** Data collection, processing, and refinement.

**Data Collection**	
Diffraction source	BL41XU, SPring8
Wavelength (Å)	1.0
Temperature (K)	100
Detector	EIGER
Crystal-to-detector distance (mm)	200.00
Rotation range per image (°)	1.0
Total rotation range (°)	280
Space group	P2_1_2_1_2_1_
a, b, c (Å)	31.34; 66.25; 96.50
α, β, γ (°)	90.0; 90.0; 90.0
Unique reflections	33,034 (1590)
Resolution range (Å)	96.5–1.50 (1.53–1.50)
Completeness (%)	99.8 (100.0)
Average redundancy	7.9 (7.2)
〈*I*/σ(*I*)〉	40.2 (4.3)
Rmeas (%)	2.7 (22.5)
CC_1/2_	100.0 (97.3)
**Refinement**	
R_fact_ (%)	17.4
R_free._ (%)	19.6
Bonds (Å)	0.01
Angles (°)	2.06
Ramachandran plot	
Most favored (%)	98.1
Allowed (%)	1.9
No. atoms	
Protein	1808
Water	158
Chromophore	23
Magnesium ions	3
Other ligands	0
B-factors (Å^2^)	
Protein	17.70
Water	31.0
Chromophore	24.6
Magnesium ions	24.60
Other ligands	0

Values in parenthesis are for the highest-resolution shell.

## Data Availability

Data are contained within the article or Appendix A.

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
