# Peer review of "The mRubyFT Protein, Genetically Encoded Blue-to-Red Fluorescent Timer"

_ijms, 2022, doi:10.3390/ijms23063208_

Round 1

Reviewer 1 Report

All my questions and suggestions were addressed and the manuscript can be published as is.

Reviewer 2 Report

the authors have addressed all of my comments

This manuscript is a resubmission of an earlier submission. The following is a list of the peer review reports and author responses from that submission.

Round 1

Reviewer 1 Report

The manuscript By Oksana M. Subach and co-authors describes a new genetically encoded fluorescent timer, mRubyFT protein. It has several advantages compared to a known, previously described FT, Fast-FT. The paper is well written, provides an in-depth investigation of the structure-property relations, and is of great significance to imaging community. It can be accepted with a minor revision.

  1. It was not clear to me why the fusion of mRubyFT with beta-actin demonstrated blue fluorescence (Fig. 4a, left panel), but the fusion  with alpha-tubulin did not. A short explanation of the difference in behavior of these two proteins would be helpful.
  2. In Fig. 4 (middle) the caption says 561ex/525/50em. As I understand, the last two numbers are for the name of the filter used. Why in Figure caption, this filter is named "561ex and 617/73em"? What emission filter was actually used?
  3. It looks like mRubyFT protein can be used in ratiometric mode. In biological studies it is very desirable to have a reference signal, because it is usually difficult, or even impossible, to measure something (in this case the time) using only on the absolute signal. Figure 2c shows a perfect chance to use blue fluorescence as a reference. Why this was not at least mentioned in the paper? In this connection, why in Fig. 4 do the authors show a combined (blue + red fluorescence), but not the ratio, red/blue. The last could be a good demonstration of the possibility to measure the time with this "timer".  
  4. Do the authors have any idea of the chemical structure of the chromophore in the blue fluorescent form? I think it would be very important to at least speculate what can this structure be?

The paper can be accepted as is, after a minor spell checking, but addressing the above questions would make it even stronger.

Reviewer 2 Report

The manuscript of Subach et al describes a novel mRuby2-derived blue-to-red protein timer by comparing it to that derived from mCherry and reported earlier. Although the two fluorescent timers are monomers with similar characteristics and spectral properties, they differ in several features such as 4-fold increase in brightness of the blue form of mRuby2FT compared to mCherry-derived probe. This might make sense, taking into account the fact that imaging at blue spectra is usually more technically difficult in routine experiments due to optical limitations of conventional microscopes. The work offers an accurate and detailed description of physical and chemical properties of a novel mRuby2FT protein. However, the data provided in this submission to demonstrate the efficiency of mRuby2FT as a timer are not entirely convincing. The authors have to address flaws in the revised manuscript.

Major

No time series of cells nor the detailed time data have been provided. This work would greatly benefit from adding this kind of information (see example in Khmelinskii et al. 2012: 10.1038/nbt.2281). Actually, only one time point was provided by fig 4. It shows both blue and red forms in actin fusion while there was only a red form in tubulin fusion at 72 hours after the transfection. No explanation of this difference has been provided either.

Fig 3c-e is very confusing. Left, it shows no cell-related red channel fluorescence of FastFT at all, whereas the mRubyFT panel shows strong red fluorescence. It seems that the summary histogram on Fig 3c is not background subtracted as the DF/F of FastFT red is only ~4 (otherwise F would be ~0, so DF/F >>4). As F of Fast FT red channel is dominated by background (i.e. by a non-biological signal), the 3.9 fold value on panel c is meaningless.

Minor

Fig 3e left – both red and blue forms are present at the initial mRubyFT picture in contrast with the in vitro data. On the other hands, there is only a blue form shown for mCherry-derived FastFT as a reference. Why this time point was selected? Were there any time points when the red form of mRubyFT was completely immature? Otherwise, the authors should explicitly state through the manuscript that their new time probe emerges as blue+red rather than blue.